# Surgical and Oncological Outcomes of En-Bloc Resection for Malignancies Invading the Thoracic Spine

**DOI:** 10.3390/jcm12010031

**Published:** 2022-12-20

**Authors:** Pierluigi Novellis, Luca Cannavò, Rosalba Lembo, Andrea Evangelista, Elisa Dieci, Veronica Maria Giudici, Giulia Veronesi, Alessandro Luzzati, Marco Alloisio, Umberto Cariboni

**Affiliations:** 1Division of Thoracic Surgery, IRCCS San Raffaele Scientific Institute, 20132 Milan, Italy; 2Division of Orthopedic Oncology and Spine Reconstructive Surgery (CCOORR), IRCCS Galeazzi Orthopedic Institute, 20161 Milan, Italy; 3Department of Anesthesia and Intensive Care, IRCCS San Raffaele Scientific Institute, 20132 Milan, Italy; 4Unit of Clinical Epidemiology, Città della Salute e della Scienza di Torino, 10126 Torino, Italy; 5Division of Thoracic Surgery, Humanitas Clinical and Research Center, 20089 Rozzano, Italy; 6Faculty of Medicine and Surgery, Vita-Salute San Raffaele University, 20132 Milan, Italy; 7Department of Biomedical Sciences, Humanitas University, 20090 Milan, Italy

**Keywords:** spine tumor, lung cancer invading spine, vertebral tumor resection, vertebral T4 NSCLC, vertebral metastases

## Abstract

Objective(s): There is still limited data in the literature concerning the survival of patients with tumors of the thoracic spine. In this study, we analyzed clinical features, perioperative and long-term outcomes in patients who underwent vertebrectomy for cancer. Furthermore, we evaluated the survival and surgical complications. Methods: We retrospectively reviewed all cases of thoracic spinal tumors treated by the same team between 1998 and 2018. We divided them into three groups according to type of tumor (primary vertebral, primary lung and metastases) and compared outcomes. For each patient, Overall Survival (OS) and Cumulative Incidence of Relapse (CIR) were estimated. Complications and survival were analyzed using a logistic model. Results: Seventy-two patients underwent thoracic spine surgery (40 in group 1, 16 in each group 2 and 3). Thirty patients died at the end of the observation at a mean follow up time of 60 months (41%). The 5-year overall survival was 72% (95% CI: 0.52–0.84), 20% (95% CI: 0.05–0.43) and 27% (95% CI: 0.05–0.56) for each group, respectively. CIR of group 3 was higher (HR 2.57, 95% CI: 1.22–5.45, *p* = 0.013). The logistic model revealed that age was related to complications (*p* = 0.04), while surgery for a type 3 tumor was related to mortality (*p* = 0.02). Conclusions: Although the cohort size was limited, primary vertebral tumors displayed the best 5-y-OS with an acceptable complications rate. The indication of surgery should be advised by a multidisciplinary team and only for selected cases. Finally, the use of a combined approach does not increase the risk of complications.

## 1. Introduction

Surgical treatment of vertebral pathologies often demands a joint approach of both thoracic and vertebral surgeons. Among the pathological processes requiring a combined approach, there are degenerative disc diseases, osteomyelitis, vertebral fractures and tumors [1,2]. Anterior exposure of vertebral bodies is a necessary step for the release and resection of the target vertebrae.

Tumors of the thoracic district with vertebral involvement can be classified into three categories: primary tumors of the spine, primary tumors of adjacent organs with infiltration of the vertebrae (especially NSCLC) and secondary tumors localized in vertebrae [3]. T4 lung tumors have traditionally been considered a contraindication to surgery due to invasion of unresectable structures. In 1975, Paulson identified vertebral invasion as a contraindication to surgical intervention [4]. During the last decades, however, surgical improvements have led to good oncological outcomes even after resection of locally advanced tumors with invasion of mediastinal structures. such as the left and right atrium (with cardiopulmonary bypass if required), tracheal carina and the superior vena cava [5]. Overall, surgical intervention provides the best local control for resect able tumors, resulting in radical resection and respecting oncologic principles. In 2002, Grunenwald reported the first results concerning the resection of T4 NSCLC with spinal invasion [6]. He observed that en-bloc resection of chest tumors with vertebrectomy, albeit technically demanding and associated with high postoperative morbidity, may have encouraging long-term survival. It could therefore be a valid option in selected patients.

Vertebral resection performed by morcellation of cortical and trabecular bone violates the margin of the surgical specimen and entails the risk of a minimal residual tumor. The development of en-bloc single or multiple vertebral body resection was facilitated by the growing experience of vertebral body replacement with cages [7]. Such a challenging operation could be evaluated for radical treatment of the pathology, as in primary spine tumors and NSCLC, or to achieve oncological and symptomatic control, as in secondary tumors. 

Previously published case series already demonstrated technically feasible approaches for en-bloc resection with vertebrectomy [6,7,8,9,10,11]. However, outcomes and long-term survival data in a large cohort are missing, as well as clear criteria for patient selection. We have evaluated clinical features, perioperative and long-term outcomes in a large cohort of total and partial en-bloc vertebrectomies for treatment of different intra-thoracic malignancies. 

The aims of this study were to assess survival in these three different tumor types and to analyze the variables impacting the 5-year mortality and complications.

## 2. Materials and Methods

### 2.1. Patients

This is a retrospective cohort study that includes patients from September 1998 to October 2018 who underwent spine en-bloc resection for primary vertebral tumors, lung cancers involving the spine or bone metastases of other malignancies. The cases were collected from two centers but were all treated by the same spine team: the Thoracic Surgery Division of Humanitas Research Hospital and the Division of Oncological Orthopedics of the Galeazzi Institute. Although the operations were performed in two different institutions, the surgical team has always dealt with these cases together. Data were extracted by clinical records. Patient′s life and death status was determined by consulting the local registry offices or, when living, by telephone contact. In that moment, patient consent was obtained. Approval for this study was provided by the Ethics Committee of the Humanitas Research Hospital (authorization n. 2216) and rated by the Galeazzi Hospital–San Raffaele Hospital Ethics committee (authorization n. 182/2019).

### 2.2. Data Variables and Outcomes

The final data set included the following variables: age at surgery, gender, comorbidities (divided in cardiological, pneumological and metabolic), ASA score [12], pre-operative and post-operative ASIA score [13], number of involved vertebrae, surgical characteristics, operative duration, type of tumor, histologic characterization, post-operative complication, histologic grade (G), tumoral residual (R) and survival data.

According to type of tumor, cases were divided into three groups: primary tumor of the vertebrae (group 1), primary tumor of the lung infiltrating the vertebrae (group 2) and secondary tumors of the vertebrae from different primary sites (group 3). The surgical approach was either single incision or combined thoracic and posterior incision. Moreover, lung resection was reported when associated with the vertebral resection. Surgery was then defined as radical (R0) when a complete tumor resection was accomplished and incomplete in case of microscopically (R1) or macroscopically (R2) residual disease. Histological grading was categorized into well-(G1), moderately (G2) and poorly differentiated (G3) cancer according to degree of architecture and cytological atypia.

### 2.3. Operative Technique

A thorough preoperative examination was carried out with each patient. Preoperative imaging typically included both computed tomography (CT) and magnetic resonance imaging (MRI), while a PET scan was routinely performed only after 2006. Cardiac and anesthesiological evaluations were required for all patients, whereas further cardiological examinations such as echocardiography, stress test and myocardial scintigraphy were required just in selected cases. Angiography was performed preoperatively to check the anterior spinal artery position. Spirometry was used only in cases in which a lung resection was pre-operatively planned. Patients sustaining neoadjuvant treatment underwent pre-operative re-staging with CT and MRI. 

In case of a combined approach, the starting position of the patient on the operating table was established to minimize the risk of spinal cord injury. A cervico-thoracic approach with manubrial split was typically used for spinal pathology involving C7 to T3 [14,15], whereas thoracotomy was adopted for T4 to T10 lesions [16,17]. The choice of the intercostal space depended on the position of the tumor, preferring one or two spaces above the targeted vertebra. The side of thoracotomy was chosen pre-operatively according to tumor infiltration, especially in case of well-differentiated tumors. Furthermore, a left thoraco-abdominal incision was used for spinal involvement between T11 and L2 to prevent liver injuries (Figure 1c) [18]. One to two chest drains were positioned and the incision was finally sutured.

The posterior incision was always performed by the orthopedists. In this case, the thoracic surgeon, who was always available in the operating room, helped in the isolation of the vertebral bodies from the adjacent structures. Once the patient was positioned prone, a midline posterior spine incision and subperiosteal dissection were executed. A “C arm” or fluoroscopy was used to confirm the target vertebral body. Laminectomies were performed one level above and one level below the invaded vertebrae. Accurate spinal cord 360° liberation was necessary for en-bloc resection. The number and location (ipsilateral or bilateral) of nerve roots that were sacrificed depended on tumor invasion and spinal artery location. Discectomies, vertebral body osteotomies or sagittal osteotomies for partial vertebrectomy were performed depending on surgical margins. Posterior spinal fixing was carried out with bilateral pedicular titanium screws (2 or 3 levels above and below the resection) and titanium or carbon fiber rods. The en-bloc specimen (comprising lung resection if present) was then extracted through the posterior incision. Finally, spinal reconstruction was completed with a vertebral body replacement cage filled with autograft bone. A prolene mesh was sometimes applied. 

### 2.4. Statistical Analysis

Categorical data are presented as frequency (percentage, %), and continuous data as mean and standard deviation or median (IQR) where appropriate. Differences in baseline characteristics were compared between complication or mortality with the use of the χ2 test or Fisher’s exact test (when appropriate) for categorical variables, while Student’s *t* test or the Mann–Whitney test were used for continuous variables if not normal distributed [19].

For each patient, overall survival (OS) was calculated from the date of surgery until death by any cause. Cumulative incidence of relapse (CIR) was also calculated in the sample. OS function was estimated by Kaplan–Maier method, whereas CIR was estimated using the method proposed by Gooley et al. [20]. 

The risk of post-operative complication and mortality were analyzed using two logistic regression models. All statistical analyses were performed using STATA (Version 16).

## 3. Results

A total of 72 patients underwent thoracic spine surgery. A total of 40 subjects underwent surgery for a primary tumor of the vertebrae (group 1), 16 subjects were affected by lung cancer infiltrating the vertebrae (group 2) and 16 subjects had a metastasis localized in the dorsal spine (group 3). The median age was 41.5 (23, 58) for group 1 tumors, 61 (53.5, 67.5) for group 2 tumors and 53 (47, 58) for group 3 tumors. Clinical characteristics of the patients are described in Table 1. Chondrosarcomas (6/40 15%) and osteosarcomas (5/40 12.5%) were the most frequently observed primary mesenchymal tumors. Cordomas were observed in 6/40 (15%) cases, representing the most frequent primitive embryonic neoplasia.

As for group 2 tumors, lung adenocarcinoma was present in 5 patients; the same number of subjects had a diagnosis of squamous cell carcinoma. A total of 10/16 (62.5%) patients were previously treated with neoadjuvant therapy, among which 4 cases (25%) received chemo-radiotherapy, 4 (25%) patients only received chemotherapy and 2 (12.5%) patients only received radiotherapy.

In the third group, the most frequent histotype was thyroid tumor metastasis (4 = 25%), whilst the second in terms of frequency was breast cancer metastasis (3 = 18.75%). In all cases of secondary tumors, vertebral localization was the only systemic metastasis. When considered the best and most radical approach, surgery was discussed with a multidisciplinary team. 

In 21 (29.1%) subjects, the demolitive intervention was performed only by posterior incision, while in 51 (70.9%) a double thoracic and posterior longitudinal approach was necessary.

The mean diameter of tumor was 58.1 mm (min 15 mm max 150 mm) for group 1, 54.6 mm (min 20 mm max 85 mm) for group 2 tumors and 48 mm (min 20 mm max 80 mm) for group 3 tumors. The mean operation time was 539 min. In 23 cases (32%), one vertebral body was involved by the tumor; in 20 cases (28%), two vertebral bodies; in 25 cases (34%), three vertebral bodies; and in 3 cases (6%), more than three vertebrae were affected. Pulmonary resection was performed in 25 patients (34%): 10 patients were subjected to wedge resection; a segmentectomy was performed in 1 patient, right upper lobectomy in 9 cases, left upper lobectomy in 2 patients and left lower lobectomy in 1 patient. A radical resection (R0) was achieved in 57 patients (79%), whereas a microscopic residue (R1) was detected in 15 patients (21%). No cases had macroscopically positive resection margins (R2). In 9 cases (12%), the tumor was well differentiated (all cases were in group 1), in 35 cases (49%) the tumor was a G2, in 27 (37%) it was a G3 and only in 1 case the tumor grading was not definable due to complete response after induction therapy. Complications were described in 41 patients (57%). 

### 3.1. Overall Survival Analysis

Thirty patients had died by the end of observation. A 30-day mortality occurred in five patients (6.9%, 95% CI: 2.3–15.5): two in group 1, two in group 2 and one in group 3. A 90-day mortality occurred in eight patients (11.1%, 95% CI: 4.9–20.7): two in group 1, four in group 2 and two in group 3. All of these patients were operated with a combined approach (thoracic and posterior). The 5-year overall survival was 72% (95% CI: 0.52–0.84), 20% (95% CI: 0.05–0.43) and 27% (95% CI: 0.05–0.56) for group 1, group 2 and group 3, respectively (Figure 2). The univariable analysis on survival showed a relation between age (*p* = 0.0048), pneumological comorbidities (*p* = 0.039), associated lung resection (*p* < 0.0001), tumor type (*p* = 0.001) and “grading” (*p* = 0.001). In the logistic model, only a type 3 tumor resection was confirmed associated (OR 6.14, *p* = 0.01) as reported in Table 2. Grading appeared to be slightly related to survival but without significance (OR 3.31, *p* = 0.08). For group 2, tumor “N status” was not related to survival but probably for the small number of observations (*p* = 0.53).

### 3.2. Secondary Objectives

The first secondary objective consisted in the observation of the cumulative incidence of relapse (CIR) in the different tumor groups. The 5-year CIR was 52% (95% CI: 0.33–0.67), 41% (95% CI: 0.17–0.63) and 72% (95% CI: 0.34–0.91) for group 1, group 2 and group 3, respectively (Figure 3). A univariate analysis was performed on complications. The analysis revealed that only the age (*p* = 0.0061), the pneumological comorbidities (*p* = 0.0036) and the Hospital stay—Days (*p* = 0.003) had a statistic correlation. According to the logistic model, age was related to complications (OR 1.03; *p* = 0.04) as reported in Appendix A.

## 4. Discussion

Vertebral resection was initially recommended for the treatment of infectious diseases of the spine such as Pott′s disease [21]. Subsequently, the indication to surgery was further extended to neoplastic pathologies.

Our series describes our team′s thirty-year experience in en-bloc resections of neoplasms with vertebral involvement. Vertebral resection was associated to a 6.9% 30-day mortality and a 11.1% 90-day mortality. Surgical procedures were extremely complex, with a 57% complication rate. This range of survival implies several considerations.

A previous research from 2015 presented a case series of 38 subjects who underwent vertebral resection for a primary vertebral tumor (group 1) [22]. Our work was extended to all cases of the team, thus creating three groups of subjects affected by neoplasms quite different from one another. Our sample was thus extremely heterogenous. Different tumors in fact have distinct epidemiological characteristics such as sex preference and age of incidence. Patients’ comorbidities and the extent of resections were also highly varied. In group 2 (NSCLC infiltrating the vertebrae), lung resection is always performed, however, this is not the case for group 1 (primary tumors of the spine).

Our primary objective was to verify and compare the overall survival of the different groups of tumors. We obtained a 5-year survival rate of 72% (95% CI: 0.52–0.84), 20% (95% CI: 0.05–0.43) and 27% in group 1, group 2 and group 3, respectively.

Over time, several studies related to vertebral resections were published. Some of them focused on clinical outcomes, but few described a thorough survival analysis [23,24]. Pettiford and colleagues observed a 5-year survival of 26% [1] in a series of 213 patients that underwent spine surgery for a heterogeneous group of pathologies. Similarly, Mody described a 5-year survival of 40.3% [7] in a series of 32 spine tumors (primary, NSCLC and secondary). In these studies, the authors did not stratify according to the different tumor types. Feng and colleagues reported a series of 16 patients with vertebral osteosarcomas and found that 516 patients (31.3%) died, but 15/16 (93%) had local or distant recurrence of the disease at a mean follow up of 42.4 months after resection [24]. In our group, albeit not statistically significant, grading seems related to survival also in the logistic model. 

For subjects of group 2, instead, the scenario is extremely different. Lung cancer remains the leading cause of cancer death [25]. Additionally, proper management of patients with extrapulmonary structures invasion in T4 disease is still controversial.

In 1996, Grunenwald described the first case of vertebral resection for NSCLC promoting its surgical feasibility [26]. In 2002, the same author described a series of 19 subjects treated with en-bloc pulmonary and vertebral bone resection for lung neoplasms infiltrating the vertebrae. The 5-year survival was 14%, which is lower than that reported in our case study [6].

A recent review of literature by pooled analysis examined 135 subjects affected by lung cancer with spinal invasion [27]. The 5-year survival of the analyzed cohort was 43%, an outcome that is not confirmed by our results. The authors found that surgical radicality (R0) was an important prognostic factor. The sample size of our study (16 subjects affected by NSCLC) did not allow us to evaluate precise prognostic factors for this subgroup of patients. Furthermore, an increase in mortality was observed in N1/N2; nonetheless, it was not considered significant. In a recent work, Dartevelle described his thirty-year experience in resection of T4 lung tumors [28], concluding that, although feasible, the selection of patients to extended surgery for lung cancer must be attentive.

In agreement with Dartevelle’s results, our small series of lung cancer with vertebral infiltration confirms a poor prognosis and high complication rate. Although our data are limited and do not allow final conclusions, the first line of treatment for T4 tumors invading the spine should be systemic treatment combined with radiotherapy. Considering the introduction of monoclonal antibodies, protein kinase inhibitors and PD1 and PDL1 check inhibitors in clinical practice [29], we believe that surgery should be addressed only after failure of medical treatments and of the other non-surgical protocols.

Among vertebral resections for metastatic bone tumors (group 3) of our sample, 5-year survival was estimated around 26%, compared to other series, in which the overall survival for these types of tumors is very poor (meaning between 8 and 12 months) [30,31].

Concerning post-operative complications, these occurred in 57% of our cases with a 30-day mortality of 6.9% and 90-day mortality of 11.1%, in line with the literature. Such high perioperative mortality and morbidity are related to the complexity of the surgical procedure. The patient must be well informed regarding the risks of the procedure and the therapeutic alternatives. Still, patients often have no valid therapeutic alternative and have already been treated with induction chemo or chemo-radiotherapy. Boriani and colleagues described a complication rate of 46.2% in a sample of 220 vertebral resections and observed that complication rate was related to multi-segmental resection of vertebrae and to the number of incisions (posterior and anterior) [32]. Pettiford described an average of 51.3% of post-operative complications, with a prevalence in cases of resection for infectious diseases that outnumbered complicated oncologic procedures [1]. Mody also described a complication rate of 56% [7]. Our series has been collected in about 20 years of activity, during which the approaches to post-operative hospitalization have changed. In relation to spinal cord problems, such as parepesis and paraplegia, it is imperative to minimize the risk of spinal trauma as much as possible. During these years, systematic angiographic study has been added to preoperative examinations in order to reduce the risk of perioperative neurological damage by neoplastic embolism. We also introduced intraoperative monitoring with Somato-Sensitive Evoked Potentials (PESS) and Motor Evoked Potentials (PEM), which gives real-time information on the activity of the spinal cord to guide surgical resection. Respiratory physiotherapy associated with early mobilization, thrombo-embolic prevention measures and good analgesia could reduce the risks of post-operative pneumonia and pulmonary thrombo-embolism. In one of two cases of aortic rupture, an aortic patch was used during surgery because of suspicion of infiltration. In cases alike, we now recommend to routinely place an aortic endoprosthesis before resection.

The limits of our research are manifold. First, it is a retrospective study that covers a very long time span, during which diagnostic and therapeutic strategies have witnessed a remarkable progression. As previously stated, the sample is also extremely heterogeneous. Lastly, the study lacks data on post-operative quality of life and functional impairment.

The strengths include a very long follow up, the homogeneous criteria for selection of candidates and the fact that the surgical technique was always applied by the same surgical team. Despite the restricted series of cases, this study allows us to observe that the treatment of primary tumors of the vertebrae, especially well-differentiated neoplasms, represents a cure from cancer in an acceptable percentage of patients. For what concerns lung cancers infiltrating the vertebrae and metastatic tumors, surgery should be discussed in a multidisciplinary context once other treatment lines have failed.

## 5. Conclusions

The best survival rate was observed in primary tumors of the vertebrae, especially in the case of well-differentiated tumors, with a 5-year survival of 72%. Subjects with non-small cell lung cancer and vertebral infiltration show a 5-year survival rate of 20%. These results reflect those reported in the literature. The surgical option must be carefully discussed. It is considered feasible only when the most innovative therapeutic approaches (such as biological therapies or immunotherapy) or other non-surgical protocols fail. Although the reduced size of the cohort does not allow definitive conclusions, the use of a combined approach is associated with a risk of complications consistent with that reported in the literature.

## Figures and Tables

**Figure 1 jcm-12-00031-f001:**
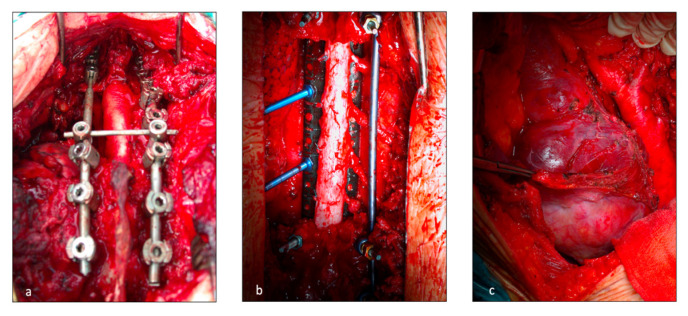
Operative field during vertebral resection and posterior spinal fixing. T (**a**) posterior release of the spinal cord at C7-D3 level; (**b**) Release of the spinal cord at D4-D9 levels; (**c**) left thoraco abdominal incision for spinal involvement at D11-L2 levels to prevent liver injuries.

**Figure 2 jcm-12-00031-f002:**
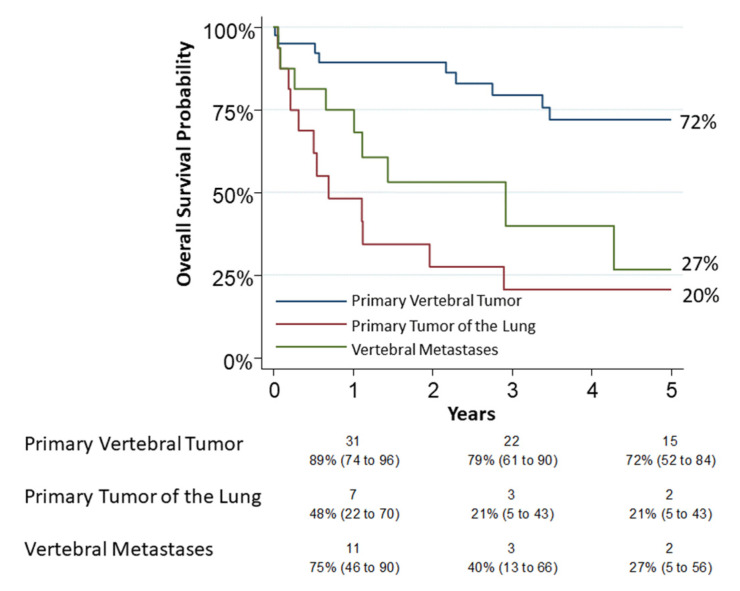
The 5-year overall survival for group 1 (primary vertebral tumors), group 2 (primary tumors of lung infiltrating spine) and group 3 (vertebral metastases) was 72% (95% CI: 0.52–0.84), 20% (95% CI: 0.05–0.43) and 27% (95% CI: 0.05–0.56), respectively. Numbers below the time axis are patients at risk and Kaplan–Maier estimates (95% CI).

**Figure 3 jcm-12-00031-f003:**
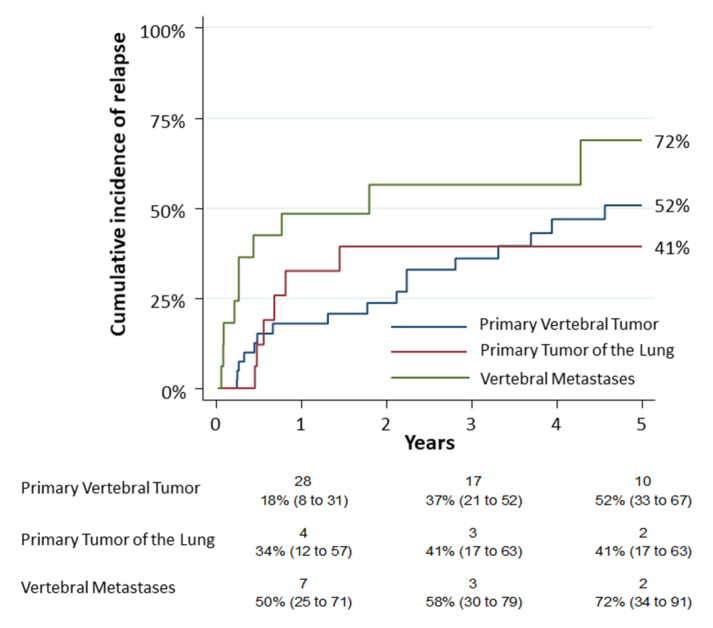
The cumulative incidence of relapse (CIR) among the different tumor groups resulted 52% (95% CI: 0.33–0.67) for primary vertebral tumors (group 1), 41% (95% CI: 0.17–0.63) for primary tumors of lung infiltrating spine (group 2) and 72% for vertebral metastases (group 3) (95% CI: 0.34–0.91). Numbers below the time axis are patients at risk and cumulative incidence estimates (95% CI).

**Table 1 jcm-12-00031-t001:** Clinical features of the sample based on type of tumor. A remarkable heterogeneity was observed in the sample. Group 1 tumor is defined as mesenchymal primary tumors, group 2 as primary tumors of the lung infiltrating the vertebrae and group 3 as vertebral metastases.

Patient Characteristics and Outcome Variables	Value (No. of Patients: 72)
Age, median–year (Mean ± SD)	48 ± 19.9
Gender, male—*n* (%)	45 (63%)
Cardiological comorbitities—*n* (%)	16 (22%)
Pneumological comorbitities—*n* (%)	10 (14%)
Metabolic comorbitities—*n* (%)	13 (18%)
Pre-Operative ASIA score	
- A—*n* (%)	2 (2.8%)
- B—*n* (%)	4 (5.6%)
- C—*n* (%)	12 (16.7%)
- D—*n* (%)	12 (16.7%)
- E—*n* (%)	42 (58.3%)
Pre-operative chemotherapy—*n* (%)	27 (38%)
Pre-operative Radiotherapy—*n* (%)	20 (28%)
No. of involved vertebrae	
- 1—*n* (%)	23 (32%)
- 2—*n* (%)	20 (28%)
- 3—*n* (%)	29 (40%)
Associated lung resection—*n* (%)	
Surgical incisions	
- Posterior Incision alone—*n* (%)	21 (29%)
- Posterior incision and thoracotomy—*n* (%)	51 (71%)
Tumors Group	
- 1—*n* (%)	40 (56%)
- 2—*n* (%)	16 (22%)
- 3—*n* (%)	16 (22%)
Grading	
- 1—*n* (%)	10 (13.9%)
- 2—*n* (%)	35 (48.6%)
- 3—*n* (%)	27 (37.5%)
Surgical time—Hours (Medians IQR)	547.5 (444.5–665.5)
ICU stay—Days (Medians IQR)	2(1–5)
Hospital stay—Days (Medians IQR)	17.5(13–22)
Patient characteristics and outcome variables	Value (No. of patients: 72)
Age, median–year (Mean ± SD)	48 ± 19.9
Gender, male—*n* (%)	45 (63%)
Cardiological comorbitities—*n* (%)	16 (22%)
Pneumological comorbitities—*n* (%)	10 14%)
Metabolic comorbitities—*n* (%)	13 (18%)
Pre-Operative ASIA score	
- A—*n* (%)	2 (2.8%)
- B—*n* (%)	4 (5.6%)
- C—*n* (%)	12 (17%)
- D—*n* (%)	12 (17%)
- E—*n* (%)	42 (58%)
Pre-operative chemotherapy—*n* (%)	27 (38%)
Pre-operative Radiotherapy—*n* (%)	20 (28%)
No. of involved vertebrae	
- 1—*n* (%)	23 (32%)
- 2—*n* (%)	20 (18%)
- 3—*n* (%)	29 (40%)
Associated lung resection—*n* (%)	
Surgical incisions	
- Posterior Incision alone—*n* (%)	21 (29%)
- Posterior incision and thoracotomy—*n* (%)	51 (71%)
Tumors Group	
- 1—*n* (%)	40 (56%)
- 2—*n* (%)	16 (22%)
- 3—*n* (%)	16 (22%)
Grading	
- 1—*n* (%)	10 (13.9%)
- 2—*n* (%)	35 (48.6%)
- 3—*n* (%)	27 (37.5%)
Surgical time—Hours (Medians IQR)	547.5 (444.5–665.5)
ICU stay—Days (Medians IQR)	2 (1–5)
Hospital stay—Days (Medians IQR)	17.5 (13–22)

**Table 2 jcm-12-00031-t002:** Clinical features of the sample based on type of tumor. A remarkable heterogeneity was observed in the sample. Group 1 tumor is defined as mesenchymal primary tumors, group 2 as primary tumors of the lung infiltrating the vertebrae and group 3 as vertebral metastases.

	Alive (60 Months)	Dead (60 Months)	*p* Value	Odds Ratio	Conf. Interval	*p* Value
Age, median–year (Mean ± SD)	42 ± 18.3	53 ± 15.6	0.0048	1.02	0.98–1.05	0.35
Gender, male—*n* (%)	20 (52%)	25 (73%)	0.067			
Cardiological comorbitities—*n* (%)	6 (15%)	10 (29%)	0.16			
Pneumological comorbitities—*n* (%)	2 (5.2%)	8 (23%)	0.039	3.09	0.38–24.76	0.12
Metabolic comorbitities—*n* (%)	4 (10%)	9 (26%)	0.12			
Pre-Operative ASIA score			0.82			
- A—*n* (%)	1 (2.6%)	1 (2.9%)				
- B—*n* (%)	1 (2.6%)	3 (8.8%)				
- C—*n* (%)	7 (18%)	5 (14%)				
- D—*n* (%)	6 (15%)	6 (17%)				
- E—*n* (%)	23 (60%)	19 (55%)				
Pre-operative chemotherapy—*n* (%)	13 (34%)	14 (41%)	0.54			
Pre-operative Radiotherapy—*n* (%)	8 (21%)	12 (35%)	0.17			
No. of involved vertebrae			0.84			
- 1—*n* (%)	11 (28%)	12 (35%)				
- 2—*n* (%)	11 (28%)	9 (26%)				
- 3—*n* (%)	16 (42%)	13 (38%)				
Associated lung resection—*n* (%)	7 (18%)	20 (58%)	<0.0001	4.07	0.69–23.80	0.18
Surgical incisions			0.32			
- Posterior Incision alone—*n* (%)	13 (34%)	8 (23%)				
- Posterior incision and thoracotomy—*n* (%)	25 (65%)	26 (76%)				
Tumors Group			0.001			
- 1—*n* (%)	29 (76%)	11 (32%)				
- 2—*n* (%)	3 (8%)	13 (38%)		1.86	0.24–14.30	0.55
- 3—*n* (%)	6 (16%)	10 (29%)		5.96	1.36–25.98	0.02
Grading			0.001	3.31	0.86–12.72	0.08
Surgical time—Hours (Medians IQR)	555 (461–660)	533.5(400–671)	0.87			
ICU stay—Days (Medians IQR)	1.5 (1–4)	2.5 (1-6)	0.12			
Hospital stay—Days (Medians IQR)	18.5 (14–22)	16.5 (13–23)	0.72			
Age, median–year (Mean ± SD)	42 ± 18.3	53 ± 15.6	0.0048	1.02	0.98–1.05	0.35
Gender, male—*n* (%)	20 (52%)	25 (73%)	0.067			
Cardiological comorbitities—*n* (%)	6 (15%)	10 (29%)	0.16			
Pneumological comorbitities—*n* (%)	2 (5.2%)	8 (23%)	0.039	3.09	0.38–24.76	0.12
Metabolic comorbitities—*n* (%)	4 (10%)	9 (26%)	0.12			
Pre-Operative ASIA score			0.82			
- A–n(%)	1 (2.6%)	1 (2.9%)				
- B—*n* (%)	1 (2.6%)	3 (8.8%)				
- C—*n* (%)	7 (18%)	5 (14%)				
- D—*n* (%)	6 (15%)	6 (17%)				
- E—*n* (%)	23 (60%)	19 (55%)				
Pre-operative chemotherapy—*n* (%)	13 (34%)	14 (41%)	0.54			
Pre-operative Radiotherapy—*n* (%)	8 (21%)	12 (35%)	0.17			
No. of involved vertebrae			0.84			
- 1—*n* (%)	11 (28%)	12 (35%)				
- 2—*n* (%)	11 (28%)	9 (26%)				
- 3—*n* (%)	16 (42%)	13 (38%)				
Associated lung resection—*n* (%)	7 (18%)	20 (58%)	<0.0001	4.07	0.69–23.80	0.18
Surgical incisions			0.32			
- Posterior Incision alone—*n* (%)	13 (34%)	8 (23%)				
- Posterior incision and thoracotomy—*n* (%)	25 (65%)	26 (76%)				
Tumors Group			0.001			
- 1—*n* (%)	29 (76%)	11 (32%)				
- 2—*n* (%)	3 (7.8%)	13 (38%)		1.86	0.24–14.30	0.55
- 3—*n* (%)	6 (15%)	10 (29%)		5.96	1.36–25.98	0.02
Grading			0.001	3.31	0.86–12.72	0.08
Surgical time—Hours (Medians IQR)	555 (461–660)	533.5 (400–671)	0.87			
ICU stay—Days (Medians IQR)	1.5 (1–4)	2.5 (1–6)	0.12			
Hospital stay—Days (Medians IQR)	18.5 (14–22)	16.5 (13–23)	0.72			

## Data Availability

Not applicable.

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
