# Peer review of "Surgical and Oncological Outcomes of En-Bloc Resection for Malignancies Invading the Thoracic Spine"

_jcm, 2022, doi:10.3390/jcm12010031_

Round 1
Reviewer 1 Report
I complement the authors on this work. Although the numbers are understandably small, their experience in surgical procedures to resect the T4 lung cancers invading vertebrae is important to be shared with the wider thoracic surgical community. While we continue to push the borders of "operability", it is important to be cognizant of the morbidity and the outcomes. The authors have outlined these very well and it is commendable.
Author Response
Thank you very much for your comments. We hope that our research will be useful to the scientific community.
Reviewer 2 Report
Looks good and interesting work, good luck.
but i have few comment:
1- in the abstract "CIR" need to be defined.
2- in table 1 the "Pre-Operative ASIA score" the sum is 100.4%? how could it be?
3- in table 1 the "No. of involved vertebrae" the sum of the groups is 90% where are the other 10%?
4- in table 2 the "Tumors Group" the sum of the groups is 98%?
5- English should be modified
6- a lot of the presented informations are already known, what is your novelty ?
Author Response
1- in the abstract "CIR" need to be defined.
Answer point 1 - I specify in the abstract both OS (overall survival) than CIR (cumulative incidence of relapse)
2- in table 1 the "Pre-Operative ASIA score" the sum is 100.4%? how could it be?
2- Thanks for the remark. We have added the first decimal to all levels of the variable, but also in this way the sum is not exactly 100 but 100.1. Also adding the second decimal would arrive at 100.1 To reach 100 we would have to add more decimals which would make the table heavy to read.
3- in table 1 the "No. of involved vertebrae" the sum of the groups is 90% where are the other 10%?
4- in table 2 the "Tumors Group" the sum of the groups is 98%?
3 - 4 - Thank you for the comments in points 3 and 4 which are typos. Thanks to your observation, we have corrected them.
5- English should be modified
5- A revised version has been attached.
6- a lot of the presented informations are already known, what is your novelty ?
Answer point 6 - Our paper outlines a series of thoracic vertebral resections. From experience gained over many years, we have understood that survival is acceptable after surgical treatment of primary tumors of the vertebrae. On the other hand, the situation is different for what concerns primary tumors of the lung and metastatic vertebral cancer. This is a novelty in a surgical scenario. A expressed in discussion the strengths of this paper is a very long follow up, the homogeneous criteria for selection of candidates and the fact that the surgical technique was always applied by the same surgical team. This study allows us to observe that the treatment of primary tumors of vertebrae, especially well-differentiated neoplasms, represents a cure from cancer in an acceptable percentage of patients. The situation is different for lung cancers infiltrating the vertebrae and metastatic tumors where surgery can be chosen only in very selected cases.

Round 2
Reviewer 2 Report
looks more interesting and good luck for the authors